# Ultrafast dynamics of heme distortion in the O$_2$-sensor of a thermophilic anaerobe bacterium

Olga N. Petrova[1,5], Byung-Kuk Yoo [1,6], Isabelle Lamarre[1], Julien Selles[2], Pierre Nioche [3,4] & Michel Negrerie[1✉]

Heme-Nitric oxide and Oxygen binding protein domains (H-NOX) are found in signaling pathways of both prokaryotes and eukaryotes and share sequence homology with soluble guanylate cyclase, the mammalian NO receptor. In bacteria, H-NOX is associated with kinase or methyl accepting chemotaxis domains. In the O$_2$-sensor of the strict anaerobe *Caldanaerobacter tengcongensis* (*Ct* H-NOX) the heme appears highly distorted after O$_2$ binding, but the role of heme distortion in allosteric transitions was not yet evidenced. Here, we measure the dynamics of the heme distortion triggered by the dissociation of diatomics from *Ct* H-NOX using transient electronic absorption spectroscopy in the picosecond to millisecond time range. We obtained a spectroscopic signature of the heme flattening upon O$_2$ dissociation. The heme distortion is immediately (<1 ps) released after O$_2$ dissociation to produce a relaxed state. This heme conformational change occurs with different proportions depending on diatomics as follows: CO < NO < O$_2$. Our time-resolved data demonstrate that the primary structural event of allostery is the heme distortion in the *Ct* H-NOX sensor, contrastingly with hemoglobin and the human NO receptor, in which the primary structural events are respectively the motion of the proximal histidine and the rupture of the iron-histidine bond.

[1] Laboratoire d'Optique et Biosciences, INSERM U1182, Ecole Polytechnique, Palaiseau, France. [2] Laboratoire de Biologie du Chloroplaste et Perception de la Lumière chez les Micro-Algues, UMR 7141 CNRS-Sorbonne Université, Institut de Biologie Physico-Chimique, Paris, France. [3] Laboratoire de Toxicité Environmentale, Cibles Thérapeutiques, Signalisation Cellulaire et Biomarqueurs, UMR S1124, Campus Saint-Germain-des-Prés, Université de Paris, Paris, France. [4] Plateforme d'Analyses Moléculaires et Structurales, BioMedTech Facilities, INSERM US36 - CNRS UMS2009, Campus Saint-Germain-des-Prés, Université de Paris, Paris, France. [5] Present address: O.N.P. Laboratoire Handicap Neuromusculaire: Physiologie, Biothérapie et Pharmacologie Appliquées, Inserm U1179, Université de Versailles Saint-Quentin-en-Yvelines, Paris, France. [6] Present address: Department of Chemistry and Chemical Engineering, California Institute of Technology, Pasadena, CA, USA. ✉email: michel.negrerie@polytechnique.edu

Dioxygen (O$_2$) and nitric oxide (NO) protein sensors are essential for bacteria to monitor changes of concentration of these diatomics and to adapt to new environmental conditions. Heme-based gas sensors, which are linked to various downstream functions, are present in numerous organisms, have evolved from diverse structural folds[1,2], and may have homologous counterparts in mammals. Particular sensors found in several bacteria species have been originally identified[3–5] based on their sequence homology with the heme domain of the mammalian NO receptor[6], namely the enzyme soluble guanylate cyclase (sGC). The sensing domain incorporates a $b$-type heme and was named heme nitric oxide/oxygen binding (H-NOX) domain because it may bind either O$_2$ or NO depending on the species, even if some H-NOX domains, including sGC, do not bind O$_2$[7,8]. The bacterial H-NOX proteins are associated with histidine kinase or diguanylate cyclase domains in the same operon or are included in full-length proteins together with a methyl-accepting chemotaxis domain[3]. A subfamily of H-NOX sensors found in facultative anaerobes (such as *Shewanella oneidensis* and *Vibrio cholerae*) are specific NO sensors that do not form a stable complex with O$_2$ similarly with sGC. The function of such bacterial H-NOX proteins is thought to regulate biofilm formation or quorum sensing signaling in a NO-dependent manner[9,10]. Another subfamily of H-NOX sensors was found in obligate anaerobes such as *Caldanaerobacter subterraneus subsp. Tengcongensis*[7,11,12], which lives optimally at high temperature (75–86 °C)[11]. This sensor (named *Ct* H-NOX hereafter) shares tertiary fold with the $\beta_1(1–194)$ sensing domain of sGC, but presents a very high affinity for O$_2$ at room temperature[13,14], contrarily to sGC, the stable oxy-complex being stabilized by the distal residue Tyr140[13,15]. Although *Ct* H-NOX sensor binds the three diatomics NO, CO, and O$_2$[13,16], it was proposed to act as an O$_2$-sensor rather than a NO-sensor[17], especially since the *Ct* H-NOX domain is fused to a methyl-accepting chemotaxis domain in a full-length protein[3]. However, this presumed function of the *Ct* H-NOX is not yet demonstrated by in vitro assays in correlation with bacteria living conditions, as performed for example in the case of the NO-transporter cytochrome *c'*[18]. The detection of O$_2$ by H-NOX sensor in obligate anaerobes should provide these bacteria with a mechanism for metabolism adaptation or O$_2$ avoidance reactions.

Heme O$_2$-sensors from other bacteria species, with different sequence and tertiary structures, undergo a structural rearrangement upon O$_2$ binding which involves the motion of particular side-chains[19]. Contrastingly, the static structures of *Ct* H-NOX in the Fe(II)-O$_2$ and Fe(III)-H$_2$O complexes indicated that heme deformations could be involved in the sensing mechanism[20–22]. Indeed, an important feature of *Ct* H-NOX is the large distortion of its heme in the Fe(II)-O$_2$ state[4,5], which departs from planarity much more than any other known heme sensors. Heme distortion is encountered in proteins with diverse folds and functions and impacts their properties[19,23,24], including oxidases, oxidoreductase, and cytochromes with $c$-type hemes, in which this distortion influences the redox potential. Besides this role, heme distortion could be involved in controling the protein conformational state through interactions between the heme and the protein. For example, a heme distortion change was observed in the bacterial CooA sensor after CO binding[25] which activates this sensor for DNA binding. The dynamics of heme distortion change was never measured to date in any protein.

Here, we address the question of the role of the heme distortion in allostery and how the heme distortion observed in steady-state crystal structures evolves with structural dynamics. We aimed at identifying and quantifying the dynamics of heme distortion triggered by the binding to and the dissociation of diatomics from the heme of the sensor *Ct* H-NOX, focussing on its interaction with O$_2$ which is crucial for this anaerobe. We recorded the nonequilibrium dynamics induced by the photodissociation of O$_2$ and its eventual rebinding to *Ct* H-NOX using transient electronic absorption spectroscopy in the picosecond to millisecond time range to detect and to disentangle the heme structural relaxation and ligand dynamics. The behavior of the sensor in the presence of O$_2$ was compared to that in the presence of NO and CO in the 1 ps to 5 ns time range. These experiments revealed a unique feature of the bacterial *Ct* H-NOX sensor: the ability of immediately changing the heme distortion upon O$_2$ release, and consequently of modulating the molecular orbitals overlap. The heme distortion (Fig. 1) which is increased upon O$_2$ binding[2,3,17] (a state that we refer to as tensed) is changed in <1 ps after O$_2$ dissociation to produce a relaxed state. The diatomic NO exerts a similar effect, but in a lower extent than O$_2$. Conversely, the tensed state of the heme is reached after O$_2$ binding in <5 μs.

## Results

Due to its high affinity, the protein was directly purified as a 6-coordinated (6-c) H-NOX–O$_2$ complex with a characteristic Soret band maximum at 416 nm, similarly with globins[26], and well-separated Q-bands at 555 and 590 nm (Supplementary Fig. S1a and Table S1). The unliganded 5-coordinated (5-c) ferrous heme, generated after thorough degasing in the presence of the reductant dithionite, has a Soret at 431.5 nm whose absorption coefficient is larger than for the oxy complex. This species was used to

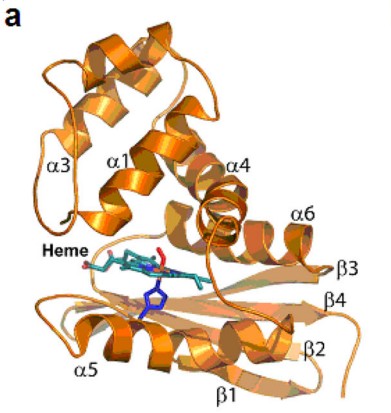
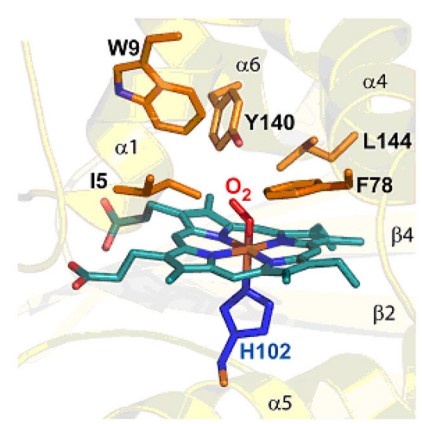

**Fig. 1 Heme distortion revealed by X-ray crystal structure. a** Overall 3D structure of the *Ct* H-NOX sensor. The heme prosthetic group is in cyan, the proximal histidine (His102) coordinating the iron in blue and the molecular oxygen in red. The figure was generated using Pymol; PDB ID: 1XBN. **b** Zoom into the heme environment and the distal amino acids surrounding the dioxygen molecule.

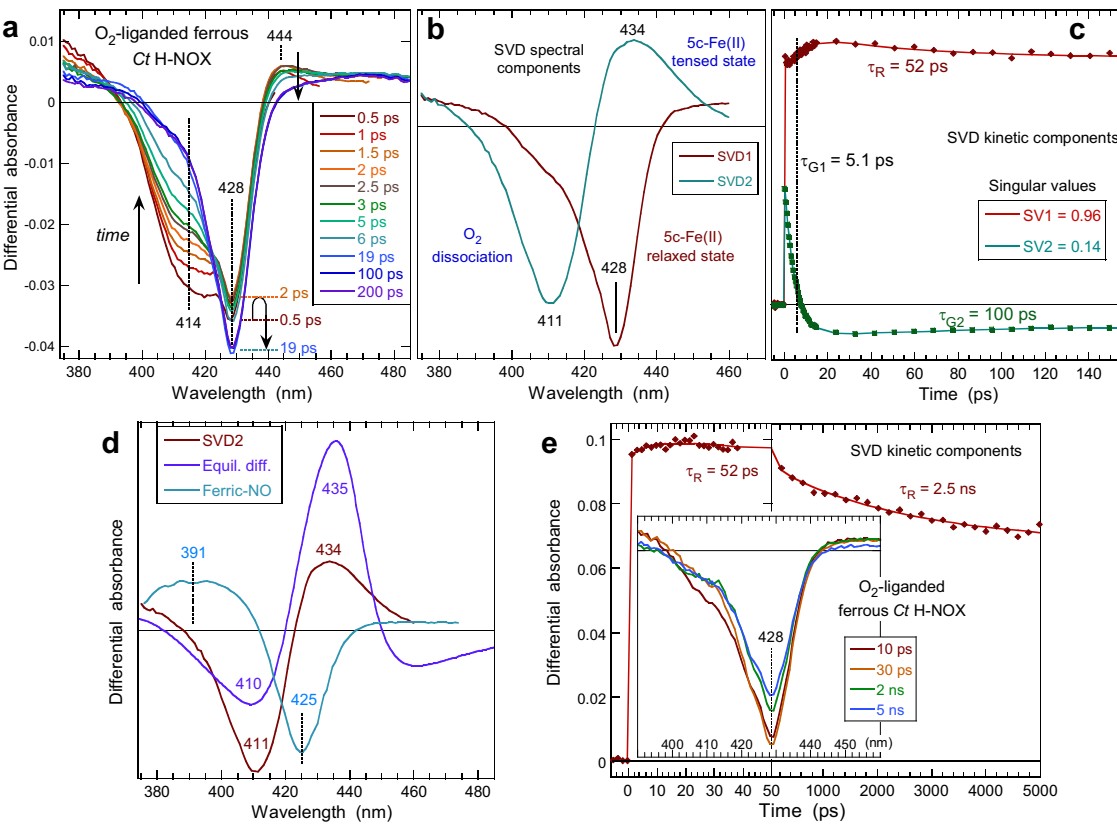

**Fig. 2 Dynamics in ferrous O$_2$-liganded *Ct* H-NOX. a** Transient difference absorption spectra (spectra at positive delay *minus* spectrum at negative delay) after the photodissociation of O$_2$ from ferrous heme at increasing time delay ($\lambda_{ex} = 564$ nm). **b** Spectral components from Singular Value Decomposition analysis of the time-wavelength data matrix of *Ct* H-NOX–O$_2$. **c** Associated SVD kinetic components fitted to the sum of two exponential terms and their time constants. The ordinate of the three panels is the difference of absorbance after *minus* before O$_2$ dissociation. SVD1 and SVD2 refer to their ranked singular values. **d** Comparison of the SVD2 spectra for the dissociation of O$_2$ with the equilibrium difference and the transient spectrum of NO dissociation from ferric *Ct* H-NOX. **e** Kinetics of the SVD1 component up to 5 ns.

prepare anaerobically the CO- and NO-liganded 6-c *Ct* H-NOX complexes. The spectrum of the *Ct* H-NOX–CO and the –NO complexes have a sharp Soret peak at 424 nm and 420 nm (Supplementary Fig. S1b and c) which have respectively larger and smaller absorption coefficients than the unliganded ferrous form. In bacterial H-NOX sensors, the proportion of 5-c–NO and 6-c–NO liganded hemes may depend upon the temperature due to the breaking of the Fe-His bond induced by NO binding, contrarily to mammalian sGC which is always 5-c–NO[27]. The difference spectrum of *Ct* H-NOX–NO, which is predominantly 6-c–NO, indeed discloses a small negative shoulder at 398 nm (Supplementary Fig. S1c) which corresponds to a very small proportion of 5-c–NO species at 20 °C.

The spectrum of ferric *Ct* H-NOX discloses a Soret band 409 nm (Supplementary Fig. S1d) characteristic of the Fe(III)–(H$_2$O) species which is confirmed by the appearance of a charge transfer band at 630 nm. The 6-c–NO ferric *Ct* H-NOX complex is characterized by a Soret band at 424.5 nm and well-defined Q-bands at 538 and 572 nm (Supplementary Fig. S1d). These values can be compared with those of nitrophorin Fe(III)–(NO) complexes (422, 533, and 569 nm) with very similar prominent Q-bands[28,29]. The equilibrium spectra of 6-c *Ct* H-NOX in the presence of coordinated diatomics are those of the samples used for time-resolved experiments.

**Heme structural change induced by dioxygen dissociation from *Ct* H-NOX.** The photodissociation of O$_2$ from the ferrous *Ct* H-NOX–O$_2$ complex produces an instantaneous absorption

decrease (bleaching) composed of a sharp negative band centered at 428 nm and a broad pronounced shoulder at 414 nm (Fig. 2a), whose amplitudes evolve in opposite direction during the first 20 ps. The transient spectra at early time delay are remarkably dissimilar from the difference between unliganded and O$_2$-liganded *Ct* H-NOX equilibrium spectra (Fig. 2d). Especially, no marked induced absorption band appeared at 435 nm in the transient spectra due to the unliganded ferrous heme, as observed for other ferrous proteins[26,30,31], but instead a very small positive band at 444–450 nm which quickly declines to a very broad and unstructured absorption spanning the 440–480 nm range (Fig. 2a) which does not correspond to the identified ferrous 5-c heme. The negative shoulder at 414 nm is assigned to the dissociation of O$_2$. It decreases faster than the main bleaching at 428 nm and evolves with different dynamics, as shown by the shift of the two isosbestic points when time elapses (400 and 440 nm), indicating that two different processes occur after O$_2$ dissociation. Importantly, the two bleachings are well resolved and evolve with different kinetics. Consequently, the respective spectral contributions associated with the two particular individual processes could be disentangled by Singular Value Decomposition (SVD)[32] analysis of the data matrix (Fig. 2b, c).

The SVD1 spectral component (larger singular value), corresponding to the immediate formation of the bleaching at 428 nm and its evolution, is assigned to the disappearance of the 5-c ferrous heme distorted state immediately after O$_2$ dissociation, as it was at equilibrium in the presence of bound O$_2$[17]. The SVD2 spectrum of photo-excited *Ct* H-NOX–O$_2$ reveals a positive induced absorption corresponding to the 5c-Fe$^{2+}$ heme

immediately dissociated, that is to say still distorted. It is assigned to $O_2$ dissociation (disappearance of 6-c $Fe^{2+}$–$O_2$) and appearance of the 5-c ferous heme, because the positive and negative absorption bands are located at the same position as in the equilibrium difference spectrum (Fig. 2d). However, the comparison between SVD2 and equilibrium difference spectra (Fig. 2d) readily reveals their dissimilarity, essentially the amplitude of the induced absorption at 435 nm. This is due to the fact that the 5-c ferrous heme is not in the same state of distortion (the SVD spectrum is the signature of a process), yet SVD analysis could retrieve the induced absorption which is contained in the data matrix, although it is not visible in the raw transient spectra (Fig. 2a), taking into consideration the entire spectral evolution. The decrease of SVD2 amplitude is thus due to geminate recombination of $O_2$. The 5-c ferrous heme changes its conformational state ultrafast (in <1 ps) after $O_2$ dissociation, which is the reason for the small absorbance at 444 nm in transient spectra (Fig. 2a), but the SVD2 spectrum has a positive induced absorption comparable to transient spectra of $O_2$ geminate rebinding in globins[26,31].

We must now interpret the SDV1 spectral component. Importantly, no induced absorption band appeared in the region 393–405 nm of SVD1 spectrum, where the 5-c ferric heme absorbs so that photo-oxidation can be discarded (see below the case of ferric-NO heme). Refering to the equilibrium difference spectra of $O_2$-bound *Ct* H-NOX (Fig. 2d), a simple $O_2$ dissociation should have produced a well-defined positive band at 435 nm. This is not the case for the SVD1 spectrum which comprises only pronounced deep bleaching at 428 nm which, together with the absence of induced absorption, implies that the heme absorption coefficient decreased, but keeping an absolute spectrum shape very close to that of the initially dissociated heme, without change of its redox or coordination state. The conclusion is that the structural state of the 5-c ferrous heme has changed immediately (<1 ps) after the dissociation of $O_2$, a ligand whose binding imposes such a constraint that the 6-c $O_2$-bound heme experiences a large distortion and immediately relaxes upon dissociation.

Another way to identify processes is to remove from a raw transient spectrum the contribution of the initial state which appears as a negative absorbance due to its disappearance (Fig. 2a). Removing the contribution of the initial $O_2$-liganded species from a difference transient spectrum implies to add its equilibrium spectrum and will make apparent only the produced states (Supplementary Fig. S2). We must note that all remaining spectral contributions are intermediate states in the present case. When only one dissociated state is produced[26,27,31], removing the initial state results in the equilibrium spectrum of the produced state. Contrarily, here, once the equilibrium spectrum of $O_2$-liganded state is added, after having removed the negative contribution at 414 nm in the transient at 2.5 ps, the second bleaching at 428 nm remains, associated with the induced absorption at 442 nm. This positive absorption is due to the produced 5-coordinate photodissociated heme, whose relaxation induces the bleaching at 428 nm and implies a decrease of its absorption coefficient.

Alternatively to SVD, the raw data kinetics were analyzed directly (Supplementary Fig. S2c and d) yielding two or three exponential components (Supplementary Table S2). The decay with time constant $\tau_G = 5.5 \pm 0.5$ ps is preponderant (A = 75 %) in the absorption range of $O_2$-liganded species and corresponds to $O_2$ geminate rebinding to the heme in both distorted and relaxed states, as analyzed below. A fast component ($\tau_{R1} = \sim 1$ ps) is important at 440 nm, where the distorted dissociated heme appeared, and is preponderant at 420 nm where it relaxes. This fastest component is thus assigned to the heme structural relaxation which is an immediate consequence of $O_2$ dissociation, and is followed by a slower relaxation ($\tau_{R2} = 53 \pm 4$ ps).

## Does photo-oxidation occur? Dissociation of NO from ferric *Ct* H-NOX.

The previous result means that the photo-dissociation of $O_2$ from *Ct* H-NOX transiently changes the ferrous heme conformation but not its redox state. Nevertheless, in order to ascertain this conclusion and to prove that 5-c ferric heme did not form, we photodissociated NO from the ferric heme, the only way to record its transient spectrum in the 5-c state since the *Ct* H-NOX ferric heme is 6-c with bound water in the steady-state, as indicated by its Soret band at 409 nm (Supplementary Fig. S1d). Indeed, at equilibrium most of the ferric hemes are 6-c with $H_2O$ (or $OH^-$) on the distal side, having a Soret band positioned at 403–410 nm, whereas the dissociated 5-c ferric heme discloses a broad Soret band centered at ~390 nm. A $H_2O$ molecule cannot diffuse from solvent and bind to the heme in the ps time range, a process which takes place in the slower µs time range[33]. After the photodissociation of NO from ferric *Ct* H-NOX, bleaching appeared immediately at 425 nm together with a broad induced absorption centered at 391 nm (Fig. 3a). They both decay simultaneously without a shift of the isosbestic points and almost vanished at 200 ps, contrastingly to the bleaching observed

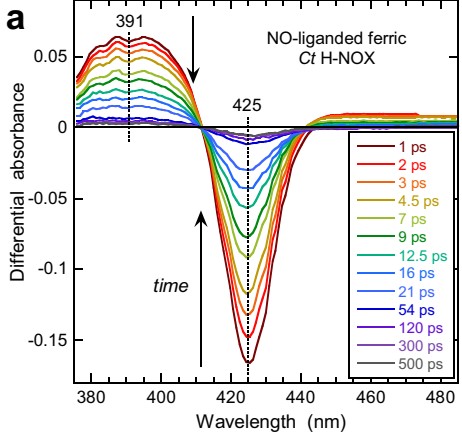
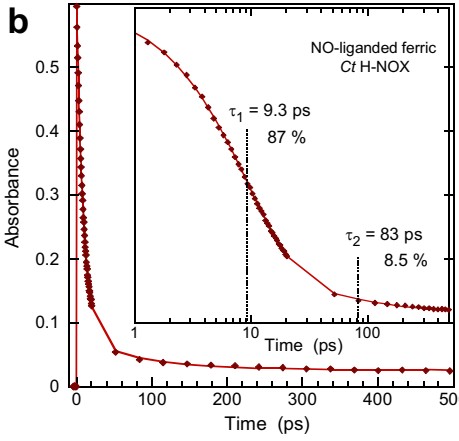

**Fig. 3 Dynamics in ferric *Ct* H-NOX sensor. a** Transient difference absorption spectra after the photodissociation of NO from ferric heme at increasing time delay ($\lambda_{ex} = 564$ nm). **b** Kinetics of NO geminate rebinding and fit to a sum of exponential terms (logarithmic scale in the inset) with their time constants. The constant term accounts for 4.5% of the amplitude.

for $O_2$-dissociated ferrous *Ct* H-NOX (Fig. 2a). The evolution of the differential spectrum is due to the geminate rebinding of NO to the ferric heme, which takes place with two-time constants ($\tau_1 = 9.3$ ps and $\tau_2 = 83$ ps, Fig. 3b) as it was observed for other ferric heme proteins interacting with NO[33,34]. The first exponential component is due to NO rebinding from within the distal heme pocket whereas the second one is due to NO rebinding from a more distant site within the protein core. The induced absorption band at 380–400 nm decreases as NO rebinds and constitutes a marker of the 5-c unliganded ferric heme, similarly to ferric sGC[35] and ferric globins[26,33]. This broad band is absent from the transient spectra of $O_2$-dissociated ferrous *Ct* H-NOX (Fig. 2a) firmly confirming that no photo-oxidation occurred in this latter species.

We further verified whether photo-oxidation could occur in 5-c ferrous unliganded *Ct* H-NOX by recording the heme excited states relaxation of this species (Supplementary Fig. S3 and Supplementary Results and Discussion). Electronic excited states decay and vibrational cooling of heme occurred, as similarly observed and well established for other ferrous heme proteins excited either in the Q-bands[27,36,37] or in the Soret band[38,39]. Importantly, the transient spectra of unliganded ferrous heme also disclose induced bleaching at 428 nm which does not reach the original baseline after 200 ps, although the heme ground state is 5-c so that no ligand can be photodissociated. Again, the absorption decrease centered at 428 nm could indicate photo-oxidation[40]. However, no induced absorption band appears in the region 380–400 nm (Fig. S3a) where the 5-c ferric heme absorbs as demonstrated for $Fe^{3+}$−NO heme (Fig. 3) and the hypothesis of photo-oxidation must be discarded in favor of a conformational change without a change of the redox state.

**Dynamics of dioxygen and structural transition of *Ct* H-NOX.** The SVD1 kinetic component first discloses a rise then a slower decay (Fig. 2c). The rise is due to the $O_2$ rebinding ($\tau_G = 5.1$ ps) which triggers the heme structural change back to its 6-c distorted and tensed conformation (inversion of the intensity trend at the minimum of the bleaching in Fig. 2a). The heme then partly relaxes with a time constant $\tau_R = 52$ ps. The SVD2 kinetic component describes $O_2$ geminate rebinding in two phases, whose fast time constant (5.1 ps) is similar to constants (4.7–7.5 ps) measured in globins[26,31,41], in FixL and DOS bacterial $O_2$-sensors[42] and in the bacterial NO-carrier cytochrome *c'* (L16A-AXCP) mutated to bind $O_2$[30], which all have tertiary folds different from *Ct* H-NOX (Supplementary Table S3). The fast kinetic phase (5.1 ps) is due to $O_2$ rebinding still in the heme pocket, in close vicinity to the heme iron (<5 Å) whereas the second phase is due to $O_2$ having migrated farther away within the protein. Similarly with myoglobin, but not with the $O_2$ sensors DOS and FixL, a slower phase took place ($\tau_2 = 100$ ps, $A_2 = 3$ %) which corresponds to rebinding of $O_2$ having diffused to a remote location in the protein. Of note, only a single $O_2$ geminate rebinding phase with a large amplitude has been measured for the bacterial $O_2$-sensors DOS and FixL (Supplementary Table S3).

When we measured the heme relaxation kinetics on a longer time scale up to 5 ns, two phases occurred with a time constant $\tau_{R1} = 52$ ps (same as in short time-range) and a second slower kinetic component with time-constant $\tau_{R2} = 2.5$ ns. They are associated with a recovery of the absorbance at 428 nm and correspond to the relaxation toward the 5-c resting state of the heme. The SVD spectral components up to 5 ns (Supplementary Fig. S4) are identical to those measured on the shorter time-scale, showing that no other process took place up to 5 ns.

The question arises as whether the reverse structural transition from relaxed to distorted heme occurs at the same time as $O_2$

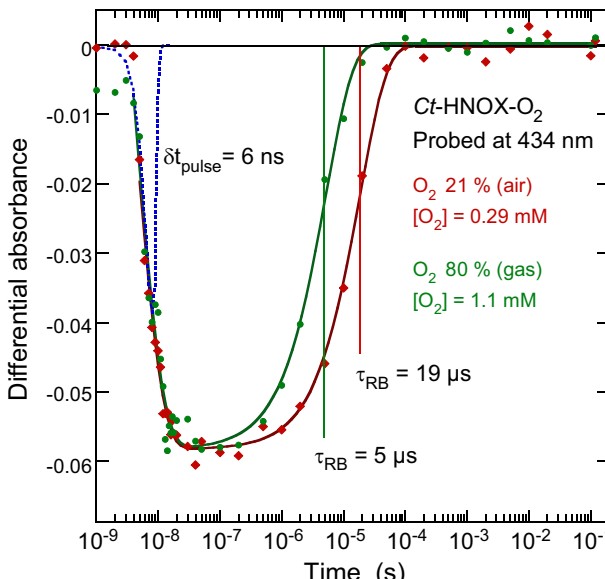

**Fig. 4 Dynamics of $O_2$ rebinding to *Ct* H-NOX in the ns to 0.1 s time scale.** The dynamics was probed at 434 nm in the presence of 21% $O_2$ and ~80% $O_2$ in the gas phase. In the first case, the sample cell was open to the air. In the second case pure $O_2$ (Air Product) was injected into the open cell. The kinetics were fitted to the sum of two exponential components. The first fast component ($\tau_1 = $ ~4 ns) corresponds to the rise of the signal (leading edge of the pulse). The second exponential is assigned to $O_2$ bimolecular rebinding from solution because its time constant depends on $O_2$ concentration ($\tau_2 = 19$ and 5 µs). The profile of the 6-ns gaussian pulse is indicated by the blue dotted curve.

bimolecular rebinding proceeds from solution. We measured the dynamics of $O_2$ rebinding after its photodissociation from ferrous *Ct* H-NOX in the ns-µs-ms time scale in the presence of 21% $O_2$ and 80% $O_2$ in gas phase (Fig. 4). The kinetics was monitored at the Soret absorption wavelength of the ferrous 5-c ferrous heme (434 nm) in order to follow its transition. An immediate negative absorption change occurred exactly as probed in the ps-ns time range, which confirms the decrease of the absorption coefficient of the 5-c ferrous heme after $O_2$ dissociation. Due to the time resolution (6 ns), the fast picosecond components of geminate rebinding of $O_2$ cannot be observed here. Importantly, we note that the measured kinetics of all 6-c $Fe^{2+}$−$O_2$ heme proteins probed at 434 nm disclose a positive induced absorbance[26,30,31,41,42] contrarily to the present case.

After the initial absorbance decay ($\tau_1 = 4$ ns) determined by the pulse shape (Gaussian curve in Fig. 4), the relaxation from heme distortion is completed within the time resolution <6 ns. Subsequently, only one exponential component could be identified, whose time constant depends on the $O_2$ concentration: $\tau_{RB} = 19$ µs at $[O_2] = 0.26$ mM and $\tau_{RB} = 5$ µs at $[O_2] = 1.1$ mM. This dependence allows to assign the transition to $O_2$ bimolecular rebinding from the solution, whose association rate can thus be calculated: $k_{on} = (1.9 \pm 0.2) \times 10^8$ $M^{-1}$ $s^{-1}$ (at 20 °C). This value is ~14 times larger than that measured by using 6-c $Fe^{2+}$−CO *Ct* H-NOX as the initial species to be photodissociated[15] (avoiding mixed kinetics due to the simultaneous presence of CO and $O_2$) and is ~4 times larger than that measured by stopped-flow ($0.43 \times 10^8$ $M^{-1}$ $s^{-1}$)[13]. This very high rate indicates a diffusion controlled $O_2$ binding from the solution and the absence of a steric barrier. Because no other transitions were observed before or after, we concluded that $O_2$ binding from the solution induces the distortion of the heme faster than 5 µs. The time constant of this

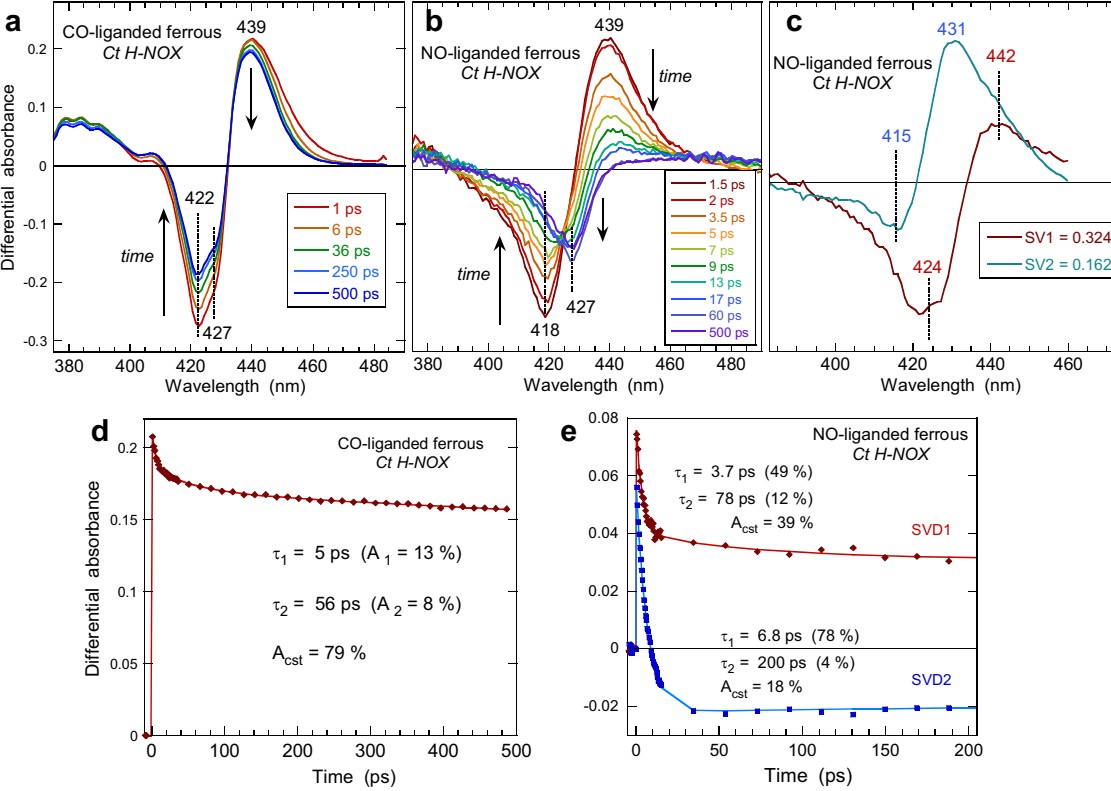

**Fig. 5 Dynamics in CO-liganded and NO-liganded ferrous *Ct* H-NOX.** Transient difference absorption spectra after the photodissociation of CO (**a**) and NO (**b**) from ferrous *Ct* H-NOX at increasing time delay ($\lambda_{ex}$ = 564 nm). (**c**) Spectral components from Singular Value Decomposition analysis of the time-wavelength data matrix of *Ct* H-NOX–NO. Associated kinetic components together with time constants for CO (**d**) and NO (**e**). Kinetics were fitted to the sum of two exponential terms with relative amplitudes $A_i$ and a constant term $A_{cst}$ (Supplementary Table S4).

allosteric transition is in the range of that measured for sGC (1–50 μs)[43].

**Heme structural distortion change induced by other diatomics.** For all $O_2$-binding proteins studied to date[26,30,31,42], a well-defined induced absorption band due to 5-c ferrous heme is present after $O_2$ dissociation and evolves with the bleaching when $O_2$ rebinds, contrarily to the present case, confirming that a fast structural change of the 5-c ferrous heme occurs in *Ct* H-NOX–$O_2$. This result led us to compare the behavior of *Ct* H-NOX–$O_2$ with the diatomics CO and NO which both bind to the ferrous heme, and to measure the transient spectra in the same conditions (Fig. 5). After the photo-dissociation of CO, intense bleaching appeared centered at 422 nm (6-c heme−CO disappears) with a shoulder at 427 nm and an induced absorption at 439 nm (5-c heme appears). The transient spectra correspond to the difference equilibrium one (Supplementary Fig. S1b), except the shoulder (427 nm) present at the same wavelength as the bleaching induced after $O_2$ dissociation (Fig. 2a) but absent in the case of ferric heme (Fig. 3a). Thus, after the dissociation of CO two different conformational states of the heme are immediately formed which remain within similar proportions during 500 ps, contrarily to $O_2$. The same relaxed heme conformation is also generated but in a much lower proportion than in the case $O_2$ as indicated by the large induced absorption remaining at 439 nm for CO but absent for $O_2$. The associated kinetics (Fig. 5d) were fitted to the sum of two exponential decays: the first one ($\tau_1$ = 5 ps) is assigned to excited states relaxation (small shift of the isosbestic point at 410 nm from 1 to 6 ps), whereas the second term ($\tau_2$ = 56 ps) is due to fast CO geminate rebinding accounting for 8% of

the total amplitude. The constant term $A_{cst}$ = 79% is due to slower bimolecular CO rebinding in the μs–ms time range. This kinetics is similar to those measured for the cognate H-NOX sensor from *Clostridium botulinum* and for guanylate cyclase[44].

The behavior of ferrous *Ct* H-NOX bound with NO (Fig. 5b) appeared intermediate between the CO- and the $O_2$-bound sensor. The photodissociation of NO induced a bleaching centered at 418 nm (6-c heme−NO disappeared) and a well-defined induced absorption at 439 nm (5-c heme appeared) similarly with CO. The transient spectrum at +1.5 ps corresponds to the difference equilibrium one (Supplementary Fig. S1c) and NO geminately rebinds to the 5-c heme in the first 20 ps. Contrarily to CO, second bleaching develops at 427 nm (absent at 1.5 ps) while the induced absorption at 439 nm, due to 5-c heme, rapidly decreases. Remarkably, there is a large shift of the isosbestic point (~430–440 nm). Similarly with $O_2$, two processes take place, namely the NO geminate rebinding and the relaxation of the distorted 5-c ferrous heme to a new conformational state. However, contrarily to $O_2$, an induced absorption appears at 439 nm due to the unrelaxed dissociated 5-c heme (Fig. 5b). Here also, we discarded the hypothesis of photo-oxidation since no induced absorption appeared at 391 nm, as observed for ferric *Ct* H-NOX–NO (Fig. 3).

The two bleaching negative parts are well separated, allowing to resolve the entangled and evolving spectral contributions of both processes by SVD analysis. The SVD1 spectrum (Fig. 5c) corresponds to the heme structural relaxation and comprises an induced absorption (442 nm) because its initial decay is slower than that of $O_2$. The SVD2 spectrum clearly corresponds to the steady-state difference (Supplementary Fig. S1c) but with a slight shift of the maximum of the positive band (436 nm). In the

transient SVD spectrum, there is a minor contribution of the 4-coordinate heme[43] due to the photo-dissociation of a small proportion of 5-c NO-heme (shoulder at 398 nm in Fig. S1c) preexisting together with 6-c heme. The presence of both species in various proportions is well known[7,8,16] for all H-NOX sensors and is negligible for *Ct* H-NOX.

Like for other heme proteins, the NO geminate rebinding appears bi-exponential (Fig. 5e) with time constants close to those observed for globins[44] (Supplementary Table S4). The heme relaxation proceeds through two exponential phases, the fastest one ($\tau_1 = 3.7$ ps) is due to vibrational excited states decay[37] whereas the second one ($\tau_{R2} = 78$ ps) is assigned to heme structural relaxation to an equilibrium state. It is similar to that measured for $O_2$. The excitation in the Q-band that we used (564 nm) allows to minimize the thermal energy to be dissipated by the heme compared to a Soret excitation[39]. The heme structural relaxation ($\tau_{R2} = 78$ ps) appears as a consequence of NO dissociation, not of electronic excitation, and could have been hardly detected in the absence of spectral resolution (which is the case when using an large open band detection)[39].

All together the present results signify that the conformation of *Ct* H-NOX ferrous heme transiently changes, but not its redox state, with an absorption spectrum different than the ground 5-c ferrous state. The lower absorption coefficient can originate from a change of orbitals overlap due to a change of heme distortion. The remarkably different evolution of the transient spectra depending on the diatomics, but with the presence of a heme relaxed species in all three cases, confirms that a fast structural change of the 5-c ferrous heme occurs for *Ct* H-NOX when $O_2$ dissociates.

## Discussion

The spectral evolution triggered by photodissociation of diatomics described here has never been observed previously in other $O_2$-binding heme proteins, for which a positive induced absorption was always measured upon $O_2$ dissociation (Supplementary Table S3), contrarily to *Ct* H-NOX. These results reveal that the dissociation of diatomics from ferrous *Ct* H-NOX induces a spontaneous structural change of the heme, but in different proportions depending on the diatomics as follows: CO < NO < $O_2$ (Supplementary Table S5). Immediately (~1 ps) after ligand dissociation, the ferrous 5-c heme macrocycle could be supposedly in the 6-c structural conformation it had before dissociation (excepting the instantaneous motion of Fe out of the heme plane)[45], but this is not the case: the transient absorption data imply that the dissociated heme adopts fastly (<1 ps) a new 5-c conformation which has a strong influence on its electronic structure, impacting its absorption coefficient. The reverse conformational change of the heme takes place upon $O_2$ binding. Let us remind that the heme in steady-state crystal structures is highly distorted in the 6-c Fe(II)-$O_2$ state[17] (tensed state) and flattened in the 5-c Fe(II) one (relaxed state). The distortion comprises both saddling and ruffling components[5,20,21]. Since in other heme proteins the $O_2$ binding does not induce a so strong distortion of the heme[5], the protein environment in *Ct* H-NOX is responsible of the unique heme distortion upon $O_2$ binding, implying that specific interactions take place between $O_2$ and distal side-chains.

The relaxation observed after NO dissociation implies that NO also induces a tensed conformation but in a lower amount. The 6-c state is even less distorted when CO is bound, but still exists and relaxes also immediately upon CO dissociation. All three ligands $O_2$, NO, and CO bound to *Ct* H-NOX induce heme distortion, but to different extents as quantified by the ratio of the two SVD components associated with heme structural relaxation and ligand rebinding (Supplementary Table S5). Remarkably, the steady-state structures bound with the diatomics reveals lower distortions in the presence of CO and NO[17]. This difference fully agrees with the relative amplitudes of SVD structural relaxation components measured for the three species, which are associated with lower energy states for CO and NO. Immediately after $O_2$ dissocation, the heme conformation is out of equilibrium, in a higher energy state, and can access the relaxed 5-c equilibrium state without energy barrier, which translates to fast picosecond kinetics, whereas a lower energy difference (also without barrier) translates to slower relaxations in the case of NO.

The heme relaxed state can be produced when the heme is electronically excited. This observation suggests that a mixture of distorted and relaxed 5-c ferrous hemes may preexist at equilibrium, which is displaced by ligand binding. Electronic excitation allows the distorted heme to relax and changes the relative populations, the initial equilibrium being restored in a time scale larger than 5 ns.

The heme distortion influences the molecular orbitals[46] and must alter the absorption coefficient. We calculated the Soret absorption spectrum of the relaxed 5-c ferrous heme from the transient spectrum after electronic excitation (Fig. 6a) by subtracting the non excited contribution from the transient spectrum

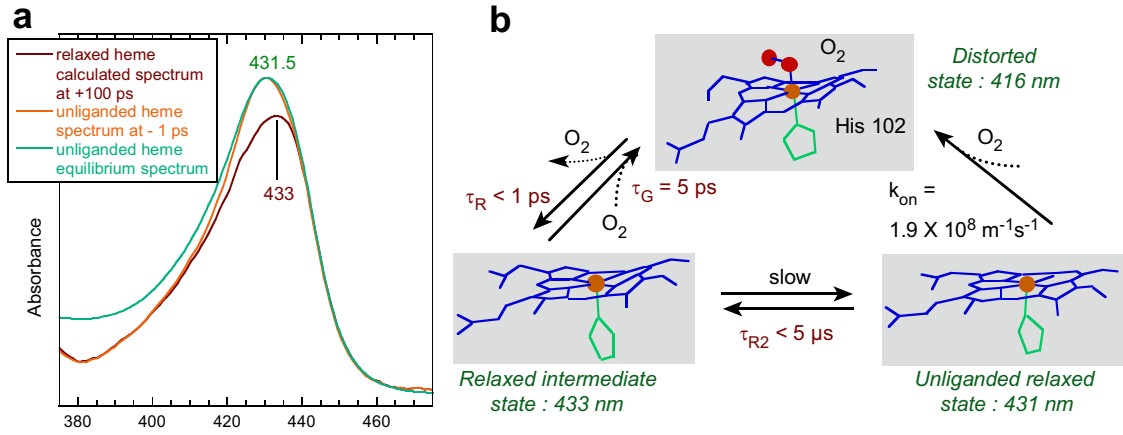

**Fig. 6 Spectroscopic fingerprint and model. a** Calculated spectrum of the relaxed 5-c ferrous heme of *Ct* H-NOX at +100 ps after $O_2$ dissociation (red) compared with the spectrum of the unliganded 5-c heme before excitation (−1 ps, orange) and with the steady-state spectrum of the same sample (green). In the latter spectrum the increased absorbance below ~400 nm is due to the reductant dithionite which is less consumed. **b** Allosteric equilibrium based on heme distortion in the *Ct* H-NOX sensor.

at +100 ps. The relaxed heme after photoexcitation has a Soret maximum shifted to 433 nm and a lower absorption coefficient compared to its unliganded equilibrium state, revealing changes in orbitals due to the change in heme distortion. Apart from the UV–visible spectrum, the heme redox potential is another electronic property which is influenced[20,22] and could be important for $Ct$ H-NOX function.

For all three ligands CO, NO, and $O_2$ the heme relaxation takes place immediately after their dissociation (picosecond) and the produced 5-c heme spectrum not only differs from that at the equilibrium 5-c unliganded state, but also differs from the 5-c spectrum would be if the heme kept the ligand-bound conformation after dissociation. Time-resolved absorption has previously shown that the spectrum of the bacterial $O_2$-sensor FixL after dissociation differs from the spectrum of the 5-c unliganded equilibrium state, being blue-shifted[42]. For FixL (with a tertiary fold different from $Ct$ H-NOX), the 5-c heme conformation immediately after ligand dissociation is the same as in the 6-c liganded state, but different from the 5-c unliganded state. Contrastingly, for $Ct$ H-NOX the heme conformation changes immediately after dissociation and is not the same as in the 5-c unliganded equilibrium state (Fig. 6). However, a common feature of FixL and $Ct$ H-NOX is the difference of heme response between the three diatomics in the order CO < NO < $O_2$, revealed by time-resolved Raman spectroscopy for FixL[47]. In the present study, this difference of reactivity is translated into the ratio of SVD amplitudes ($\gamma_{SVD}$) of the transient spectral components (Supplementary Table S5). However, the allosteric mechanisms in FixL and $Ct$ H-NOX sensors are based on different properties and different interactions between heme and protein, which reflect their adaptation to different environments.

To summarize concisely, our time-resolved spectroscopic data demonstrates that heme distortion is released in <1 ps after $O_2$ detachment. This dynamics of the heme (so far not described) controls the allosteric equilibrium in the $Ct$ H-NOX sensor. The dynamics of heme distortion appears as another mechanism for triggering allostery in heme proteins, in contrast with hemoglobin and guanylate cyclase, in which the primary structural events are respectively the motion of the proximal histidine linked to the heme and the rupture of the iron-histidine bond.

## Methods

**Preparation of the samples**. The sample of bacterial H-NOX domain from *Caldanaerobacter tencongensis* comprising the first 191 amino-acids ($Ct$ H-NOX), homologous to the heme domain of sGC was overexpressed[4]. The protein was purified in the $O_2$-liganded ferrous state (as verified by its absorption spectrum, Supplementary Fig. S1a), kept at −80 °C in triethanolmaine (TEA) buffer supplemented with 5% glycerol and was thawed immediately before use.

For all steady-state and time-resolved absorption measurements, quartz cells with an optical path of 1 mm were always used (Hellma, 110-QX). The steady-state spectrometer is a Shimadzu UV-1700. All measurements were performed at 20 °C. The absorbance of the samples was in the range 0.5–0.8 at the Soret maximum for 1-mm path length.

To measure the spectra of ferrous $Ct$ H-NOX coordinated with $O_2$ we used the protein as purified. An aliquot of $Ct$ H-NOX (120 μL at 50 μM) in a buffer (50 mM Tris-HCl pH 7.5) was placed in a cell kept in equilibrium with air, corresponding to 290 μM of $O_2$ in the aqueous phase at 20 °C. We have verified that its spectrum is identical to that obtained by reducing the ferric protein (see below) and exposing it to air. The absorption spectrum of the samples was verified at each step of the preparation and after each time-resolved experiment.

**Preparation of ferrous $Ct$ H-NOX liganded with NO or CO**. An aliquot of $Ct$ H-NOX (120 μL at 50 μM) in a buffer (50 mM Tris-HCl pH 7.5) was degased directly in the spectroscopic cell sealed with a rubber stopper. Thorough degasing was obtained by means of four successive cycles of vacuum (0.3 mbar) and purging with pure argon (1.3 bar) for 10 min between each cycle. Then, 10 μL of degased sodium dithionite ($Na_2S_2O_4$) were added with a syringe (1 mM final concentration) and the cell was heated 5 min at 50 °C to facilitate $O_2$ dissociation. Dithionite eliminated remaining traces of $O_2$. The fully reduced unliganded $Ct$ H-NOX (Soret

band maximum at 432 nm, Supplementary Fig. S1) was nitrosylated by introducing 10% NO gas (diluted in nitrogen) in the vacuumed cell, at a final pressure of ~1.3 bar, yielding 200 μM NO in the aqueous phase at 20 °C. Rigorous airtightness of the cell was ensured by putting vacuum grease at the top of the first stopper and then the second stopper in silicone. For preparing the CO-liganded protein the same procedure was used, but introducing 100% CO gas in the cell.

**Preparation of ferric $Ct$ H-NOX**. The as-prepared sample of $Ct$ H-NOX-$O_2$ (~40 μM, 120 μL) was placed in a spectroscopic cell and thoroughly degased with repeated cycles of vacuuming and purging with argon. Then the sample, still $O_2$-liganded, was oxidized by mild heating (65 °C for 20 min). The Soret appeared at 409 nm with the presence of a small absorption band centered at 630 nm (Supplementary Fig. S1d). The cell was again degased and NO gas (at 10% in $N_2$ yielding 200 μM of NO in solution) was introduced into the cell. The Soret band shifted to 424.5 nm with well-defined Q-bands at 538–572 nm.

**Picosecond to nanosecond time-resolved absorption spectroscopy**. The photodissociation of CO, NO, and $O_2$ was achieved with an excitation pulse at 564 nm, in the Q-band absorption of the heme, whose duration was ~50 fs with a repetition rate of 30 Hz. The energy of one pump pulse (~0.1 μJ) does not damage the sample, which was continuously moved perpendicularly to the beams to ensure sample renewal between pulses. The probe pulse (<50 nJ, 375–500 nm) was part of a broad band continuum generated by focusing a 620-nm pulse (~3 μJ, 50 fs) in a 1-cm water cell. Both pump and probe pulses (respective beam waists of ~150 μm and ~100 μm) were focused and overlapped in the sample cell. The optical path length of the cell was 1 mm. After the sample cell, the probe beam is directed into a monochromator (model H25, Jobin Yvon) equipped with a nitrogen-cooled charge-coupled detector (EG&G Princeton Applied Research). The transient Soret absorption was recorded as a function of time delay between pump and probe pulses. Up to 40 scans were recorded and averaged with a dwell time of 1 s for each individual transient spectrum.

Transient spectra and kinetics were simultaneously recorded to generate a time-wavelength data matrix $\Delta A(\lambda, t)$. Analysis of the data was performed by singular value decomposition (SVD) of this time-wavelength matrix[32,37] which allows the separation of entangled transient spectral components. In brief, the experimental data matrix $\Delta A(\lambda, t)$ composed of differential absorption spectra at different time delays was decomposed according to

$$\Delta A(\lambda, t) = \Delta A^{SVD}(\lambda) \cdot S \cdot K^{SVD}(t)$$

giving the matrix $\Delta A^{SVD}(\lambda)$ of orthogonal spectral components and the matrix $K^{SVD}(t)$ of associated kinetics, weighted by the singular values $S_i$ (elements of the diagonal matrix S)[32]. This procedure (home-written software) allows the identification of processes associated with particular spectral components. The SVD kinetic components were fitted to the sum of a minimum number of exponential components. Alternatively, kinetics were also performed at particular wavelengths of the raw data matrix. The temperature of the samples was 20 °C for all experiments.

A reference pulse (same energy and spectrum as the probe pulse) is recorded simultaneously with the probe, allowing to calculate the absolute absorbance for each pixel and time delay, and thus the absolute transient spectra. There is always a contribution from the non photo-excited species, which can be removed by subtracting the spectrum before excitation, with a weighting coefficient, from the spectrum at a given time delay. We could thus obtain the pure absolute spectrum of the relaxed heme (Fig. 6a) after photo-excitation, whereas the Figs. 2–5 present difference transient spectra to clearly show the change.

**Nanosecond to second time-resolved absorption spectroscopy**. For time-resolved absorption in the extended time-range nanosecond to second, we have used the home-built spectrophotometer at Institut de Biologie Physico-Chimique (Paris). This system comprises two lasers which are electronically synchronized[48]. The dissociating pulse is provided by the second harmonic (532 nm) of a Nd/YAG laser and has a duration of 6 ns. The probing pulses (duration 5 ns) were provided by a tunable optical parametric oscillator pumped by the third harmonic of another Nd/YAG laser. The sample cell compartment and light collection design allowed us to record signal variations of absorbance $\Delta A/A$ as low as $10^{-5}$. The pump and probe pulses (10 μJ) were not focused into the sample cell but spread over the surface of the sample (~1 cm²) by means of a bundle of optical fibers[48]. The kinetics of differential absorption changes were probed at particular wavelengths by tuning the optical parametric oscillator. Up to twelve scans were averaged for each kinetics. The time delay after the dissociating pulse was changed linearly from 1 to 30 ns, then was changed with a logarithmic progression from 30 ns to 1 s. The kinetics at a particular wavelength was globally fitted to the sum of a minimum number of exponential components. The temperature of the samples was 20 °C.

## Data availability

The authors declare that all data supporting the findings of this study are included in the main manuscript file or Supplementary Information or are available from the corresponding author upon request.

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

## Acknowledgements
P.N. was supported by an Avenir starting grant, Region Ile-de-France under a CODDIM contract, an INSERM-Paris Descartes University Chaire mixte. O.N.P. was supported by an "Initiative Doctorale Interdisciplinaire" PhD fellowship from Université Paris-Saclay.

## Author contributions
M.N. conceived and designed the research. P.N. produced and purified the protein. I.L. and O.N.P. performed biochemical analysis. O.N.P., B.-K.Y., and J.S. performed time-resolved measurements. M.N., O.N.P., and B.-K.Y. analyzed the data. M.N. and O.N.P. prepared the figures and wrote the manuscript. All authors approved the final version of the manuscript.

## Competing interests
The authors declare no competing interests.
