## [Peer Review File · Communications Chemistry]

Reviewers' comments:

Reviewer #1 (Remarks to the Author):

The manuscript entitled „Ultrafast dynamics of heme distortion in the O₂-sensor from a thermophilic anaerobe bacteria" by Olga N. Petrova et al., focuses on the heme distortion after O₂ binding and on the role of the heme distortion in allosteric transitions of the strict anaerobe *Caldanaerobacter tengcongensis* O₂-sensor (Ct H-NOX).

Strength

The presented time-resolved spectroscopic data demonstrate that heme distortion in the strict anaerobe *Caldanaerobacter tengcongensis* O₂-sensor (Ct H-NOX) is released in less than 1 ps after O₂ detachment. The authors suggest that this dynamics of the heme (so far not described) controls the allosteric equilibrium in the Ct H-NOX sensor. The dynamics of heme distortion appears as another mechanism for triggering allostery in heme proteins, in contrast with hemoglobin and guanylate cyclase, in which the primary structural events are respectively the motion of the proximal histidine linked to the heme and the rupture of the iron-histidine bond. The experiments are well designed and the results are solid and reliable. The manuscript is well written and readable. I think that this interesting work would certainly advance our understanding of the molecular mechanism of the heme-based oxygen sensors. I strongly recommend its publication in the Communications Chemistry.

Concerns that need to be addressed

1. References

a. I feel that citation is a little bit biased. In the Introduction about heme-based O₂ and NO sensor (Page 3), I would suggest the authors to cite suitable reviews and to discuss a little further more based on reviews such as *J. Biol. Chem.* 288, 27702 (2013): Heme-based O₂ sensors or *Chem. Rev.* 115, 6491 (2015): Heme-based gas (O₂, NO, CO) sensors.

b. It would be better to cite and discuss relevant studies of other heme-based O₂ sensors especially with the globin-coupled sensing domain (for example, YddV, AfGcHK instead of comparing your results with hemoglobin or other not oxygen sensing globins) probably in Discussion (Pages 14-17). Accordingly, Tables in Supplementary will be changed [especially those using the same technique by other groups, for example, *Biochemistry* 54, 5017 (2015), *J. Phys. Chem. Lett.* 7, 69 (2016) and so on].

2. Typos and abbreviations

a. I miss a list of abbreviations. It was sometimes difficult to read the manuscript since some abbreviations complicated understanding of the text. Please, explain all abbreviations in the text at their first place of experience (for example SVD should be explain on the page 6 not at the end of the manuscript on the page 18, IBPC on page 18, CCD on page 18, delOD/OD on page 19, OPO on page 19).

b. The figure on page 8 should be marked as Fig. 3.

c. The title - either „from a thermophilic anaerobe bacterium" or „from thermophilic anaerobe bacteria"

d. The sentence at the top of page 14 misses a verb „Contrarily to CO, a second bleaching develops at 427 nm (absent at 1.5 ps) while the induced absorption at 439 nm, due to 5-c heme." Probably „disappears" was removed?

e. The sentence on the page 14 „However, the unrelaxed 5-c heme appeared first after NO

dissociation, as shown by the absorption at 439 nm, which was not the case for O₂ⁿ is illustrated in the Figure 4 not in the Figure 5 as is stated.

3. Figure 5 and inflection points

The second exponential curve is assigned to O₂ bimolecular rebinding from solution. The time constant depends on O₂ concentration ($\tau_2 = 19$ and $5 \mu\text{s}$). I wonder how the time constants were calculated from the curves. According to my eyes, the inflection points of the green and brown curves should be moved to the lower time points. Please, could you explain the method of the time constant calculation from Fig. 5?

4. Figure 7a and calculated spectrum

In the Fig. 7a there is a calculated spectrum of the relaxed 5-c ferrous heme of Ct H-NOX + 10 ps after O₂ dissociation shown in red. Please, explain how the spectrum was calculated.

5. Vacuum and dithionite

a. How big portion of the proteins was denaturated by the vacuuming and the dithionite presence?

b. The ferrous Ct H-NOX samples liganded with NO or CO were prepared in the presence of dithionite, while the ferrous Ct H-NOX samples liganded by O₂ was without dithionite. Could there be some interference of the dithionite? I believe that dithionite degrades O₂. You have verified that the Fe(II)O₂ spectrum is identical to that obtained by reducing the ferric protein and exposing it to air, but I wonder more about the presence of dithionite in the other two samples. How did you reduce the ferric heme without dithionite? Did you remove dithionite after reduction? Please explain this in the text.

6. Oligomeric state of Ct H-NOX

It was shown that oligomeric state of some heme-based O₂ sensors is important for their function and signal transduction. Please, can you discuss the oligomeric state of Ct H-NOX which, in addition to dynamics of heme distortion, appears as another mechanism for triggering allostery in the protein? Could the heme distortion in the Ct H-NOX sensing domain influence its oligomeric state? Please see J. Biol. Chem. 295, 1587 (2020) and J. Biol. Chem. 292, 20921 (2017) for your reference.

If those concerns are adequately addressed, the manuscript could be significantly improved and strengthened.

Reviewer #2 (Remarks to the Author):

The manuscript written by O.N. Petrova and coworkers is part of the large family of articles facing challenging, fundamental questions: how proteins works at atomic scale ? In other words how structural and dynamical properties might be used to obtain certain functions.

In particular the authors have studied an O₂-sensor protein (CtHNOX) that they believe uses significant distortion of the HEME group as allosteric control contrary to more standard histidine involving changes observed in well studies HEME containing proteins.

As such this work deserves great interest.

The manuscript is clearly the outcome of serious, thorough experimental work that I appreciate being written up and published.

There are though few important points that prevent me from giving green light for publication. These major points need to be carefully addressed I think.

- P1: SVD components are orthonormal

The authors discuss in great length the SVD components but they seem to assign them the importance reserved to the spectra of physical components. SVD components are simply orthonormal ($\sum_i a_i b_i = 0$, $\sum_i a_i a_i = 1$). Despite obvious it seems that this is considered by the authors; not even the fact that two components are enough to describe the data is shown/discussed.

For example at the end of page 8 the authors write "The respective contributions of the two negative bands were disentangled by SVD analysis of the data matrix, which yielded two main spectral components (Fig. 4b), each assigned to a particular individual process."

Unfortunately the impossibility of assigning SVD components to individual process is the single most limiting drawback of SVD.

The authors need to find a better way to proceed. Maybe fitting with known components (static O₂-ligated, hot-component from shortest delay of ferrous-unligated?, etc.).

Importantly the comparison with the static spectrum is not very good (possibly simply because of the orthonormality conditions that "distorts" the SVD components away from the spectra of physical components.

- P2: evidence for HEM distortion

I might have missed it but I don't see any clear cut argument proving that this is observed. What ultimately make them state that HEM distortion is at play? Not the shape of the differential spectrum (no comparison with theory), nor the position of a given peak. The fact that the a similar SVD1 differential is observed for unligated-ferrous and O₂-ferrous?

The only argument seems to be the information provided in fig7 but as argument it seems a bit weak. Information on the calculations are not given and the distorted bound state is not calculated.

- P3: Related to P2; the author state (end of pag 6, referring to unligated ferrous C_tHNOX):

"Consequently, the only possible inference is that the resulting state of the ferrous 5-c heme has changed so that its absorption spectrum is close to that of its initial ferrous state but with a lower absorption coefficient, which can be obtained through a change of orbitals overlap due to a change of heme distortion." As I mentioned for P2, I think this argument is too weak and further experimental-theoretical inputs should be added

Few other minor points the author could consider:

- The two sections describing fig 2 and 3 (unligated ferrous and ferric C_tHNOX) could be moved in the SI to discuss (after the nicely written introduction) directly the most-important points of the paper

- Nowhere the experimental details are given with sufficient details (energy density of excitation, concentration, etc. for each experiment)

- Similarly there is an "abuse" of arbitrary units for the differential absorbance. I would clearly state for each figure what is the magnitude of the expected changes. This could be also compared with the pump intensity and the static differential to make sure that differences of the right order of magnitudes are measured.

- I think to make comparison simpler each differential absorbance spectrum should have a small top panel showing the ground state spectrum

I hope they authors will provide argument to strengthen their important outcome, I look forward to reading a new version

Reviewer #3 (Remarks to the Author):

This paper reports on the heme dynamics upon photodissociation of diatomic ligands (O₂, NO, CO) in the H-NOX (heme nitric oxide/oxygen binding) domain of the O₂-sensor from a thermophilic anaerobe bacteria. The H-NOX hemes are known to be highly distorted in the O₂-bound state, which may be relevant to the protein function. Thus, the authors focused on tracking a heme conformational change after the photolysis by using time-resolved UV-visible spectroscopy. As one of the major results, the authors found an ultrafast (unusual) absorbance decay of heme Soret peak at 428 nm upon the photolysis, and assigned it to rapid heme conformational relaxation from the highly distorted form. The SVD analysis was applied to identify the 428-nm absorbance decay process. This work is placed in proper modern context of ultrafast spectroscopy, and the observed spectral behavior is quite new, unique to a distorted H-NOX heme. Thus, I recommend publication after minor revision as commented below.

Comments:

- (1) In many other heme proteins, O₂ binding does not induce the ruffling and saddling heme distortion, which implies that not O₂ binding itself but the specific protein environment of H-NOX surrounding the heme must induce the heme distortion. The authors should state the role of the protein moiety in the heme distortion more clearly based on the H-NOX crystal structures, and discuss why only O₂-photodissociation can cause flattening of the heme within only 1 ps. This will help the readers to understand the current finding and authors' view about the relationships between the heme distortion, O₂-binding/dissociation, and protein conformational change.
- (2) The authors used "allostery" in the manuscript many times, but authors' intention is somewhat vague. What event could be relevant to the heme distortion as an "allosteric" effect?
- (3) The SVD1 spectrum for the NO photo-dissociated sample has a distinct positive peak at 442 nm (Fig. 6c), which is somewhat different from the cases for non-liganded and O₂-bound samples. Some comments should be given about this point. In Fig. 6b, the 442 nm positive contribution seems not so significant. Generally, SVD cannot yield the unique solution. Any linear combination of the calculated spectra and time traces could be a solution, to my understanding. Therefore, careful analysis is necessary, and thus this reviewer requires some comments on the above point.
- (4) For all figures presenting spectra, signal intensities should be given with the scale of y-axis. This should not be omitted.
- (5) In Methods section, the pump photon density in the spectral measurements should be written, which is important to ensure that the 428 nm signal is not due to sample photo-damage. Details of SVD analysis should be also given in Method section.

Reviewers' comments:

Reviewer #1 (Remarks to the Author):

The manuscript entitled „Ultrafast dynamics of heme distortion in the O₂-sensor from a thermophilic anaerobe bacteria" by Olga N. Petrova et al., focuses on the heme distortion after O₂ binding and on the role of the heme distortion in allosteric transitions of the strict anaerobe *Caldanaerobacter tengcongensis* O₂-sensor (Ct H-NOX).

Strength

The presented time-resolved spectroscopic data demonstrate that heme distortion in the strict anaerobe *Caldanaerobacter tengcongensis* O₂-sensor (Ct H-NOX) is released in less than 1 ps after O₂ detachment. The authors suggest that this dynamics of the heme (so far not described) controls the allosteric equilibrium in the Ct H-NOX sensor. The dynamics of heme distortion appears as another mechanism for triggering allostery in heme proteins, in contrast with hemoglobin and guanylate cyclase, in which the primary structural events are respectively the motion of the proximal histidine linked to the heme and the rupture of the iron-histidine bond. The experiments are well designed and the results are solid and reliable. The manuscript is well written and readable. I think that this interesting work would certainly advance our understanding of the molecular mechanism of the heme-based oxygen sensors. I strongly recommend its publication in the *Communications Chemistry*.

Concerns that need to be addressed

1. References

a. I feel that citation is a little bit biased. In the Introduction about heme-based O₂ and NO sensor (Page 3), I would suggest the authors to cite suitable reviews and to discuss a little further more based on reviews such as *J. Biol. Chem.* 288, 27702 (2013): Heme-based O₂ sensors or *Chem. Rev.* 115, 6491 (2015): Heme-based gas (O₂, NO, CO) sensors.

We agree and have added these two references (Refs. 1 and 2 in the revised manuscript).

b. It would be better to cite and discuss relevant studies of other heme-based O₂ sensors especially with the globin-coupled sensing domain (for example, YddV, AfGeHK instead of comparing your results with hemoglobin or other not oxygen sensing globins) probably in Discussion (Pages 14-17). Accordingly, Tables in Supplementary will be changed [especially those using the same technique by other groups, for example, *Biochemistry* 54, 5017 (2015), *J. Phys. Chem. Lett.* 7, 69 (2016) and so on].

As required these two references has been added in Supplementary Tables and the discussion was improved.

2. Typos and abbreviations

a. I miss a list of abbreviations. It was sometimes difficult to read the manuscript since some abbreviations complicated understanding of the text. Please, explain all abbreviations in the text at their first place of experience (for example SVD should be explain on the page 6 not at

the end of the manuscript on the page 18, IBPC on page 18, CCD on page 18, delOD/OD on page 19, OPO on page 19).

We added a list of abbreviations. In addition, OD (optical density) has been replaced by absorbance (A), which is more rigorous.

b. The figure on page 8 should be marked as Fig. 3.

c. The title - either „from a thermophilic anaerobe bacterium” or „from thermophilic anaerobe bacteria”

d. The sentence at the top of page 14 misses a verb „Contrarily to CO, a second bleaching develops at 427 nm (absent at 1.5 ps) while the induced absorption at 439 nm, due to 5-c heme.“ Probably „disappears“ was removed?

e. The sentence on the page 14 „However, the unrelaxed 5-c heme appeared first after NO dissociation, as shown by the absorption at 439 nm, which was not the case for O₂“ is illustrated in the Figure 4 not in the Figure 5 as is stated.

All these errors and typos have been corrected.

3. Figure 5 and inflection points

The second exponential curve is assigned to O₂ bimolecular rebinding from solution. The time constant depends on O₂ concentration ($\tau_2 = 19$ and $5 \mu\text{s}$). I wonder how the time constants were calculated from the curves. According to my eyes, the inflection points of the green and brown curves should be moved to the lower time points. Please, could you explain the method of the time constant calculation from Fig. 5?

The time constants were calculated from the fitted exponential decays $A(t) = A_0 \times e^{-(t/\tau_i)}$. Thus, at $t = \tau_i$, the initial amplitude is divided by 2.718. The inflexion point ($A_0/2$) is usually taken as an approximation of τ_i .

4. Figure 7a and calculated spectrum

In the Fig. 7a there is a calculated spectrum of the relaxed 5-c ferrous heme of Ct H-NOX + 10 ps after O₂ dissociation shown in red. Please, explain how the spectrum was calculated.

We agree that we should have better explained this point. This is detailed in the revised version of the methods section.

In brief, transient spectra are presented as difference (spectrum(t) – spectrum before excitation) in the previous figures to clearly show the change. However we always record absolute transient spectra (there is a reference beam to calculate $\log(I/I_0)$ for each pixel, thus each wavelength). Since there is always a contribution from the non-photodissociated species, one can subtract the ground state species (negative time) with a weighting coefficient in order to obtain the pure absolute spectrum of the photodissociated species.

5. Vacuum and dithionite

a. How big portion of the proteins was denaturated by the vacuuming and the dithionite presence?

The proteins are never denatured by vacuuming. As for dithionite, denaturing may come from a decrease of the pH. However the pH is controlled by an appropriate buffer and we used the smallest possible concentration of dithionite. The absorption spectrum of the samples is verified at each step of their preparation.

b. The ferrous *Ct* H-NOX samples liganded with NO or CO were prepared in the presence of dithionite, while the ferrous *Ct* H-NOX samples liganded by O₂ was without dithionite. Could there be some interference of the dithionite? I believe that dithionite degrades O₂. You have verified that the Fe(II)O₂ spectrum is identical to that obtained by reducing the ferric protein and exposing it to air, but I wonder more about the presence of dithionite in the other two samples. How did you reduce the ferric heme without dithionite? Did you remove dithionite after reduction? Please explain this in the text.

The only way to reduce the ferric heme is to use a reductant. In the case of *Ct* H-NOX it is impossible to remove dithionite after reduction (either by HPLC or filtration) because of the very high O₂ affinity. To prepare the ferrous samples liganded with NO or CO we used the smallest possible concentration of dithionite, like we did for all other heme proteins that we studied so far. We never observed interference of dithionite on the dynamics and processes, for any protein. The only effect of its presence is a very slow consumption of NO (several hours), but the spectrum of the samples was always verified after experiments to guarantee that the sample is still ferrous and NO-liganded.

6. Oligomeric state of *Ct* H-NOX

It was shown that oligomeric state of some heme-based O₂ sensors is important for their function and signal transduction. Please, can you discuss the oligomeric state of *Ct* H-NOX which, in addition to dynamics of heme distortion, appears as another mechanism for triggering allostery in the protein? Could the heme distortion in the *Ct* H-NOX sensing domain influence its oligomeric state? Please see *J. Biol. Chem.* 295, 1587 (2020) and *J. Biol. Chem.* 292, 20921 (2017) for your reference.

We agree that dimerization is important for signal transduction in the case of the bacterial heme-based oxygen sensor *Af*GcHK. However, in the case of *Ct* H-NOX sensor, and generally for all H-NOX sensors, dimerization has never been observed or reported so far to control or to alter their functioning. The heme distortion in *Ct* H-NOX has been linked to ligand binding by several methodologies (absorption and Raman spectroscopy, X-ray diffraction ...) but not linked to its oligomeric state.

If those concerns are adequately addressed, the manuscript could be significantly improved and strengthened.

Reviewer #2 (Remarks to the Author):

The manuscript written by O.N. Petrova and coworkers is part of the large family of articles facing challenging, fundamental questions: how proteins works at atomic scale ? In other words how structural and dynamical properties might be used to obtain certain functions.

In particular the authors have studied an O₂-sensor protein (CtHNOX) that they believe uses significant distortion of the HEME group as allosteric control contrary to more standard histidine involving changes observed in well studies HEME containing proteins.

As such this work deserves great interest.

The manuscript is clearly the outcome of serious, thorough experimental work that I appreciate being written up and published.

There are though few important points that prevent me from giving green light for publication. These major points need to be carefully addressed I think.

- P1: SVD components are orthonormal

The authors discuss in great length the SVD components but they seem to assign them the importance reserved to the spectra of physical components. SVD components are simply orthonormal ($\sum_i a_i b_i = 0$, $\sum_i a_i a_i = 1$). Despite obvious it seems that this is considered by the authors; not even the fact that two components are enough to describe the data is shown/discussed.

For example at the end of page 8 the authors write "The respective contributions of the two negative bands were disentangled by SVD analysis of the data matrix, which yielded two main spectral components (Fig. 4b), each assigned to a particular individual process."

Unfortunately the impossibility of assigning SVD components to individual process is the single most limiting drawback of SVD.

The authors need to find a better way to proceed. Maybe fitting with known components (static O₂-ligated, hot-component from shortest delay of ferrous-unligated?, etc.).

Importantly the comparison with the static spectrum is not very good (possibly simply because of the orthonormality conditions that "distorts" the SVD components away from the spectra of physical components).

We agree with the Reviewer. Indeed, Singular Value Decomposition is designed to provide orthogonal components contained in a data matrix. The interpretation of SVD components has been a long debate for decades. Although they are separated by matrix calculation, the SVD spectral components originate from raw experimental data and, as such, have a physical meaning, provided they can be clearly identified. We agree that "assigning SVD components to individual process is the single most limiting drawback of SVD" but this is far from impossible. The key of SVD analysis is the understanding of the origin of the differences between a SVD spectrum and the spectrum of the real underlying process. Several phenomena, independent from each other and not always present, contribute to the differences between a SVD spectrum and a process spectrum. In our case these phenomena include group velocity dispersion of the probe pulse (which may slightly distort the transient spectra), the

presence of non biological processes (vibronic relaxation) and the simultaneous presence of several states of the protein (which may add a minor component).

Thanks to our long experience for using SVD analysis on many proteins, we can identify and estimate these phenomena (which have small singular value) and associate SVD spectra with individual processes which convey biophysical and biochemical signification (which have large singular value).

Concretely, for O₂-liganded Ct H-NOX the SVD2 spectrum (Fig. 2b of the revised version) identifies a positive induced absorption corresponding to the 5c-Fe²⁺ heme immediately dissociated. It may seem that "the comparison with the static spectrum (Fig. 2d) is not very good", essentially the amplitude of the induced absorption at 435 nm. This is normal because the 5-coordinate ferrous heme is not in the same state of distortion and the transition is ultrafast, yet the SVD analysis could retrieve this component which is contained in the entire data set, although this induced transient absorption is not visible in the raw transient spectra (Fig. 2a) and could not have been revealed by other means. This induced absorption in the SVD2 component is retrieved because the kinetic evolution of the spectral components is taken into account, in other words, there is more information.

In some previous studies we have chosen to calculate also the "Decay-Associated Spectrum" (for example in Ref. 37) which represents the spectrum of a particular fitted decay. This is useful when processes are spectrally entangled with very close kinetic constants (the case of cytochrome c), but in the case of O₂-liganded Ct H-NOX we found that only one time-constant (5.1 ps) appeared simultaneously in the two SVD kinetics, meaning that O₂ can rebind to the heme in both the relaxed and distorted states.

We would like to emphasize that the two bleaching negative parts are well separated, allowing to identify the two distinct processes, both for O₂ and NO-liganded Ct H-NOX. For NO, the SVD2 spectrum (Fig. 5c) corresponds to the steady-state difference (Supplementary Fig. 1c). The origin of the slight difference between both is readily identified: in the SVD spectrum there is a minor contribution of the 4-coordinate heme (shoulder at 398 nm in Fig. S1c) due to the photo-dissociation of a small proportion of 5-c NO-heme preexisting together with 6-c heme. The presence of both species (in various proportions) is normal for all H-NOX sensors and is negligible for Ct H-NOX at 20 °C. Again, the contributions from the two processes (NO rebinding from 6c-NO species and distorted heme relaxation) could be separated from each other by SVD because they are associated with well resolved different bleachings and the comparison between the SVD spectrum of NO rebinding and the equilibrium difference is quite good here.

Furthermore, the "hot-component from shortest delay" indeed exists, but is minor since we use an excitation wavelength in the Q-band (564 nm) and not in the Soret band, so that most of the energy is used for photodissociation, not for populating heme excited states as in the case of unliganded protein. It is separated in small SVD components.

To summarize, our experience about SVD with numerous proteins taught us that phenomena or processes that may "distort the SVD components away from the spectra of physical components" can be identified and taken into account, but in these experiments the "orthonormality conditions" do not distort the SVD components (although it could be the case for different kinds of experiments).

This point was not correctly described in the previous version and we modified the discussion accordingly. In Fig. 2d we replaced the transient spectrum at -1 ps with the SVD2 spectrum.

- P2: evidence for HEM distortion

I might have missed it but I don't see any clear cut argument proving that this is observed. What ultimately make them state that HEM distortion is at play ? Not the shape of the differential spectrum (no comparison with theory), nor the position of a given peak. The fact that the a similar SVD1 differential is observed for unligated-ferrous and O₂-ferrous ?

The only argument seems to be the information provided in fig 7 but as argument it seems a bit weak. Information on the calculations are not given and the distorted bound state is not calculated.

Our data do not demonstrate the existence of heme distortion *stricto sensu*, but that the non planar geometry changes. Our data show the existence of a different heme structural conformation immediately after O₂ dissociation and demonstrate the ultrafast relaxation from one structural form to the other. The heme distortion has been evidenced by X-ray crystal structures (Refs. cited). We have reorganized the text and rewritten several parts to make the rationale clearer.

We attempted to calculate the intermediate spectrum by DFT. However, no functionals used could satisfactorily reproduce the steady-state spectrum of the unliganded heme and distorted heme, so that the theoretical result does not appear reliable.

- P3: Related to P2; the author state (end of pag 6, referring to unligated ferrous Ct HNOX): "Consequently, the only possible inference is that the resulting state of the ferrous 5-c heme has changed so that its absorption spectrum is close to that of its initial ferrous state but with a lower absorption coefficient, which can be obtained through a change of orbitals overlap due to a change of heme distortion." As I mentioned for P2, I think this argument is too weak and further experimental-theoretical inputs should be added.

We have rewritten this part to make clearer the rationale (moved on page 9, after Fig. 3). The data obtained with the unligated ferrous Ct HNOX demonstrate that the photo-excitation in itself cannot induce oxidation and that the electronic excited state is favorable to the formation of a less distorted heme conformation. The change of orbitals overlap is a hypothesis to explain why the absorption coefficient decreased.

Few other minor points the author could consider:

- The two sections describing fig 2 and 3 (unligated ferrous and ferric Ct HNOX) could be moved in the SI to discuss (after the nicely written introduction) directly the most-important points of the paper.

We agree and have placed Fig. 4 (now Fig. 2) immediately after the introduction, moving the previous Fig. 2 about unligated ferrous and ferric Ct HNOX in the SI file. We kept the Fig. 3 about ferric heme-NO in the main text (now after ferrous-O₂) because it seems important to us to show the transient spectrum of the ferric heme, which is use to discard the hypothesis of photo-excitation.

We thank the Reviewer for his/her appreciation on the introduction.

- Nowhere the experimental details are given with sufficient details (energy density of excitation, concentration, etc. for each experiment).

We have added these informations in the revised version of manuscript.

- Similarly there is an "abuse" of arbitrary units for the differential absorbance. I would clearly state for each figure what is the magnitude of the expected changes. This could be also compared with the pump intensity and the static differential to make sure that differences of the right order of magnitudes are measured.

We agree and we have added the values of absorbance in all spectra.

- I think to make comparison simpler each differential absorbance spectrum should have a small top panel showing the ground state spectrum.

I hope they authors will provide argument to strengthen their important outcome, I look forward to reading a new version.

Reviewer #3 (Remarks to the Author):

This paper reports on the heme dynamics upon photodissociation of diatomic ligands (O₂, NO, CO) in the H-NOX (heme nitric oxide/oxygen binding) domain of the O₂-sensor from a thermophilic anaerobe bacteria. The H-NOX hemes are known to be highly distorted in the O₂-bound state, which may be relevant to the protein function. Thus, the authors focused on tracking a heme conformational change after the photolysis by using time-resolved UV-visible spectroscopy. As one of the major results, the authors found an ultrafast (unusual) absorbance decay of heme Soret peak at 428 nm upon the photolysis, and assigned it to rapid heme conformational relaxation from the highly distorted form. The SVD analysis was applied to identify the 428-nm absorbance decay process. This work is placed in proper modern context of ultrafast spectroscopy, and the observed spectral behavior is quite new, unique to a distorted H-NOX heme. Thus, I recommend publication after minor revision as commented below.

Comments:

(1) In many other heme proteins, O₂ binding does not induce the ruffling and saddling heme distortion, which implies that not O₂ binding itself but the specific protein environment of H-NOX surrounding the heme must induce the heme distortion. The authors should state the role of the protein moiety in the heme distortion more clearly based on the H-NOX crystal structures, and discuss why only O₂-photodissociation can cause flattening of the heme within only 1 ps. This will help the readers to understand the current finding and authors' view about the relationships between the heme distortion, O₂-binding/dissociation, and protein conformational change.

We agree. Although we mentioned this point in the first paragraph of the discussion, the comparison to other O₂-binding heme proteins was not enough emphasized. We improved the discussion to specifically address this point.

(2) The authors used "allostery" in the manuscript many times, but authors' intention is somewhat vague. What event could be relevant to the heme distortion as an "allosteric" effect?

In the obligate anaerobe *Caldanaerobacter Tengcongensis*, the sensor *Ct* H-NOX is fused to a methyl accepting chemotaxis domain in a full length protein. The detection of O₂ by the sensor domain activates chemotaxis through an allosteric change of the linked domain.

The conformational change of the *Ct* H-NOX sensor upon O₂ binding was also evidenced by orthogonal assay with the kinase domain from *Vibrio cholerae* (Ref. 17) whose activity is regulated by *Ct* H-NOX. When the sensor is unliganded or liganded with NO or CO, the activity of the kinase is decreased. When it is liganded with O₂ its activity is not decreased. The binding of O₂, and thus the heme distortion, changes the overall conformation of the *Ct* H-NOX sensor. Consequently, heme distortion has a direct allosteric effect on the kinase activity.

(3) The SVD1 spectrum for the NO photo-dissociated sample has a distinct positive peak at 442 nm (Fig. 6c), which is somewhat different from the cases for non-liganded and O₂-bound samples. Some comments should be given about this point.

In the case of non-liganded sample, since there is no photo-dissociation, more energy from the absorbed photon is available to electronically excite the heme. In that case the transient spectra contain a larger contribution of the excited states relaxation and the SVD1 spectral component appears somewhat different. It possesses a maximum at 447 nm (whose omitted label was added in the revised Fig. S2) compared to 442 nm for ferrous NO-liganded heme. The somewhat different spectral shape is readily explained by the difference in mechanism : in the first case the relaxed heme is produced from unliganded heme in electronic excited state, whereas in the latter case it is produced from the photo-dissociated distorted heme. For ferrous O₂-liganded heme, it is produced from a much more distorted heme.

We have added comments about this point in the revised manuscript (page 14).

In Fig. 6b, the 442 nm positive contribution seems not so significant. Generally, SVD cannot yield the unique solution. Any linear combination of the calculated spectra and time traces could be a solution, to my understanding. Therefore, careful analysis is necessary, and thus this reviewer requires some comments on the above point.

It is true that sometimes SVD does not yield a unique solution. However, in this precise case, the two bleaching negative parts are well separated and allow to identify two distinct processes. Their respective SVD spectra can be unambiguously separated. The positive band at 442 nm in SVD1 (Fig. 5c, formerly Fig. 6) corresponds to a decay different than at 431 nm in SVD2. Albeit it is not seen in the raw data (Fig. 5b), the entire positive band contains contributions from the two processes which could be separated from each other by SVD,

because they are associated with well resolved bleachings. [See also answer to comment P1 of Reviewer #2].

(4) For all figures presenting spectra, signal intensities should be given with the scale of y-axis. This should not be omitted.

We agree and have modified the figures as required. However we cannot add y-values for some panels when we compare the shape of SVD and equilibrium difference spectra (for example Fig. 2d).

(5) In Methods section, the pump photon density in the spectral measurements should be written, which is important to ensure that the 428 nm signal is not due to sample photo-damage. Details of SVD analysis should be also given in Method section.

We have indicated the energy of both pulses in the Methods section and added details about the SVD analysis. The steady-state absorption spectrum of the proteins is systematically verified after each measurement to ensure that the protein is not denatured. We have employed this time-resolved method to the study of tens of heme proteins over the years, and we never observed photo-damage when using these low energy pulses. In the second time-resolved set-up (5 ns to 1 s) the pump and probe pulses (10 μ J) were not focused into the sample cell but spread over the surface of the sample (~ 1 cm²) with a bundle of optical fibers, ensuring a low energy density and a high signal/noise ratio.

Reviewers' comments:

Reviewer #1 (Remarks to the Author):

All previous concerns were adequately addressed and the manuscript was significantly improved and strengthened. Now I recommend accepting it.

Reviewer #2 (Remarks to the Author):

The authors have addressed some of my concerns but unfortunately not the most important one (that has been raised also by reviewer #3).

Nobody doubts that extended knowledge of the authors. Yet sometime lack of time or motivation can push all of us to take shortcuts we may regret one day.

The role of the reviewing system is to help avoiding such issues.

Therefore, before accepting the article, I request that an alternative analysis be performed. As the authors say the kinetics are relatively simple so it should be essentially straightforward.

What I suggest to avoid having to change the manuscript and the main figures (that as an author myself I know how annoying it can be) is to perform such analysis, hopefully confirming the authors conclusions and add a section with a concise comparison in the supplementary material.

Reviewer #3 (Remarks to the Author):

I find the revised version sufficiently improved and suitable for publication.

Ultrafast dynamics of heme distortion in the O₂-sensor from a thermophilic anaerobe bacterium

Olga N. Petrova, Byung-Kuk Yoo, Isabelle Lamarre, Julien Selles, Pierre Nioche and Michel Negrier

Response to Reviewer # 2

The authors have addressed some of my concerns but unfortunately not the most important one (that has been raised also by reviewer #3).

Nobody doubts that extended knowledge of the authors. Yet sometime lack of time or motivation can push all of us to take shortcuts we may regret one day.

The role of the reviewing system is to help avoiding such issues.

Therefore, before accepting the article, I request that an alternative analysis be performed. As the authors say the kinetics are relatively simple so it should be essentially straightforward.

What I suggest to avoid having to change the manuscript and the main figures (that as an author myself I know how annoying it can be) is to perform such analysis, hopefully confirming the authors conclusions and add a section with a concise comparison in the supplementary material.

We took the Reviewer's suggestion into consideration. We have added a paragraph in the manuscript and a Figure in Supplementary material to address this point.

Beside SVD analysis, we could use another way to identify processes by removing from a raw difference transient spectrum the contribution of the initial state, which appears as a negative absorbance due to its disappearance. Removing the contribution of the initial O₂-liganded species from the difference transient spectrum necessitates to add its equilibrium spectrum to make apparent only the produced states, as we show in Supplementary Fig. S2b. Then all remaining spectral contributions are produced intermediate states.

Consequently, after having added the equilibrium spectrum of initial O₂-liganded state, (removing its negative contribution at 414 nm) the second bleaching at 428 nm remains in the transient at 2.5 ps, associated with the induced absorption at 442 nm. This positive absorption is due to the immediately produced 5-coordinate photodissociated heme, whose relaxation induced the bleaching at 428 nm, implying a decrease of the absorption coefficient.

Alternatively to SVD, we analyzed directly the raw data kinetics (Supplementary Fig. S2c and d). The decay with time constant $\tau_G = 5.5 \pm 0.5$ ps is preponderant (75 %) in the absorption range of the O₂-liganded species and corresponds to O₂ geminate rebinding to the heme in both distorted and relaxed states. Moreover, a fast component ($\tau = \sim 1$ ps) is present and important at 440 nm, where the distorted dissociated heme immediately appeared, and is preponderant at 420 nm where it relaxes. This fastest component can thus be assigned to the heme structural relaxation which is an immediate consequence of O₂ dissociation.

REVIEWERS' COMMENTS:

Reviewer #2 (Remarks to the Author):

[Editorial note: the reviewer offered no further comments to the authors.]